# The effects of body dissatisfaction, sleep duration, and exercise habits on the mental health of university students in southern China during COVID-19

Bang Liu[1], Xuesheng Liu[2], Lin Zou[1], Jun Hu[1], Yueming Wang[1], Ming Hao[1]*

1 School of Public Health and Health Management, Gannan Medical University, Ganzhou City, Jiangxi Province, China, 2 Liaoning Province Center for Disease Control and Prevention, Institute for the Prevention and Control of Infectious and Communicable Diseases, Shenyang City, Liaoning Province, China

* hm48922200@yahoo.co.jp

**Data Availability Statement:** Raw data are available from the Open Science Framework (OSF) with accession number DOI 10.17605/OSF.IO/B65Z8.

## Abstract

Following the outbreak of COVID-19 at the end of 2019, universities around the world adopted a closed management model and various restrictive measures intended to reduce human contact and control the spread of the disease. Such measures have had a profound impact on university students, with a marked increase in depression-related psychological disorders. However, little is known about the specific status and factors influencing the impact of the pandemic on student mental health. Addressing this gap, this study examines the body dissatisfaction, physical activity, and sleep of university students during the pandemic, and uses their levels of depression to provide a theoretical basis for the development of mental health interventions for university students in the post-epidemic era. To achieve this, a total of 1,258 university students were randomly recruited for this cross-sectional study. Collected data included respondents' anthropometric measurements, body dissatisfaction levels, dietary habits, sleep status, physical activity levels, and depression levels. The overall detection rate of depression was 25.4%, with higher levels of depression among women. Multiple regression analysis showed that the PSQI score ($\beta$ = 1.768, P < 0.01) and physical activity scores ($\beta$ = -0.048, P < 0.01) were significant predictors of depression in men, while the PSQI score ($\beta$ = 1.743, P < 0.01) and body dissatisfaction scores ($\beta$ = 0.917, P < 0.01) were significant predictors of depression in women. Mental health problems were prevalent among university students during the COVID-19 pandemic. Results indicate the possibility of alleviating depression among university students by improving their body dissatisfaction, physical activity, and sleep. However, as this study was limited to Ganzhou City, it is challenging to extrapolate the findings to other populations. As this was a cross-sectional study, a causal relationship between depression levels and lifestyle habits cannot be determined.

**Funding:** This study was supported by the Starting Research Fund from the Gannan Medical University, QD202121 Research start-up funds for high-level talents, Project number: 2044 and Ganzhou city federation of social science 2021-028-0538. There was no additional external funding received for this study. The funders had no role in study design, data collection and analysis, decision to publish, or preparation of the manuscript.

**Competing interests:** The authors have declared that no competing interests exist.

## 1. Introduction

The first known outbreak of the highly contagious novel coronavirus, COVID-19, occurred in late 2019 [1]. The World Health Organization (WHO) subsequently declared COVID-19 a public health emergency of international concern on January 30, 2020 [2]. To prevent the spread of the disease, countries and regions around the world implemented various containment models—seriously disrupting daily life and causing significant mental stress [3, 4]. Such measures saw a dramatic reduction in interpersonal communication and significant increase in feelings of isolation, impacting mental health [5]. Indeed, several studies have demonstrated the association between the long-term nature of the COVID-19 pandemic and a decline in mental health, with excessive restraint having a particularly negative impact on mental health [6]. Research indicates that the number of people with major depression and anxiety disorders increased by 28% and 26% [7], respectively, in the first year of COVID-19 alone. Indeed, depression increased by 27.6%—that is, some 53 million people, of whom over 35 million were women [7]. Overall, the pandemic was found to have the greatest impact on women and young people [7].

Depression is one of the most common mental health disorders. With symptoms including prolonged low mood and depression, decreased interest in life, a lack of motivation, and loss of energy, depression can cause a variety of adverse consequences, such as anxiety and sleep disorders [8]. At the global level, the rate of depression detected in the general population was significant before the COVID-19 outbreak [9, 10]. Indeed, according to data released by the WHO on the global prevalence of depression, the number of people with depression reached 322 million in 2018 [11], second only to heart and respiratory disease as a common health condition that causes physical disability [11].

Certainly, the results of a 2016 survey revealed that 18.5% of American adults had experienced mental health problems while 4.2% had suffered a serious mental illness [12]. Meanwhile, in the United Kingdom, studies have documented a steady decline in mental health among young people over time, with the prevalence of mental health issues increasing from 14–22% to 19–32% between 1991 and 2018 [13]. A novel form of coronavirus pneumonia, COVID-19 exacerbated mental health problems in the general population [5]. For instance, a cross-sectional survey of 2,031 college students at a US university during the 2020 pandemic revealed that up to 48.14% of respondents exhibited moderate to severe levels of depression [14]. Additionally, over 33% of the sample of 254 undergraduate students enrolled at a UK university during the 2020 epidemic could be categorized as clinically depressed—a significant increase in depression levels and decrease in well-being compared to pre-epidemic findings [15].

Several factors have been found to impact mental health and well-being, including body image, sleep, and physical activity. Body image is a multidimensional construct referring to an individual's perceptions and attitudes toward their body and appearance [16]. Body image can be defined as the mental image that a person has of their body, including body size and shape, as well as their feelings, thoughts, and behaviors toward these features [16]. Body dissatisfaction occurs when there is a discrepancy between an individual's ideal and actual body shape and is thought to be closely related to emotions, thoughts, and behaviors [17]. Body dissatisfaction is common among university students [18]. In the context of the pandemic, the relationship between body dissatisfaction and the development of depression remains unclear.

Depression is also frequently accompanied by sleep disorders, with the latter considered both a risk factor and a symptom of depression [19]. Defined as barriers to entering or maintaining sleep, sleep disorders can be divided into short-term and long-term sleep disorders [20]. Short-term sleep disorders can affect a person's memory and concentration, while long-

term sleep disorders can weaken the immune system, facilitating a number of potential physical and mental illnesses, including depression [20]. Research shows that prolonged sleep latency, sleep disturbance, and early awakening are closely related to depression, and that psychoeducation is a simple and effective way to treat insomnia in conjunction with conventional treatment [21]. In respect to the third factor, studies have shown that rapid economic and sociocultural development have resulted in a decline in physical activity levels and rise in sedentary behavior [22]. Simply put, the more people exercise, the lower their depression scores. Regular exercise is considered to be beneficial for both physical and mental health, and thus has a positive effect on depression [23].

The transition from high school to university is a special time when university students have to make numerous important decisions and face increasing academic and employment pressures [24]. Amid such significant psychological pressures, students are more likely to experience depression [24]. The impact of such pressures on students' social development cannot be ignored. Existing research had generally focused on improving exercise and sleep to alleviate depression, although this approach is relatively narrow and has inconsistent outcomes [25]. However, there are relatively few studies on alleviating depression by improving the level of body dissatisfaction of college students. Accordingly, this study investigates the relationship between body image, sleep, physical activity, and the depression levels of university students in China. In doing so, this study provides a new perspective for improving depression among college students, with its findings expected to aid such initiatives going forward. Exploring whether improving body dissatisfaction can alleviate depression in university students, this study provides a scientific basis for the early prevention of and intervention in depression among this cohort.

## 2. Method

This cross-sectional study surveyed a randomly selected comprehensive university in Ganzhou City, Jiangxi Province, southern China, from October to December 2021. Through outreach to study rooms and dormitories, this study randomly invited a total of 1,355 university students enrolled in the selected university to participate in this study. The average age of respondents was 19 ± 1.03 years. Respondents provided anthropometric measurements and completed a questionnaire measuring depression, sleep, and physical activity. The questionnaire also collected respondents' age, grade, major, whether they were an only child, ideal weight, and average monthly living expenses. Completed questionnaires were received from a total of 1,258 respondents, for an effective return rate of 93%.

### 2.1 Anthropometric measurements

Height was measured using a height ruler (Seca 213, Germany) with an accuracy of 0.1 cm, while weight (0.1 kg), body fat percentage, and muscle mass (kg) were measured using a body composition analyzer (Tanita BC-610, Japan). Body mass index ($kg/m^2$) was calculated using the height and weight measurements.

### 2.2 Questionnaires

**2.2.1 Level of depression.** This study assessed depression severity using the validated Chinese version of the Self-rating depression scale (SDS), originally developed by Zung in 1965 [26], which is appropriate for investigating the degree of depression in young people and has been used more widely in China. The SDS scale comprises 20 items divided into four levels of scoring; the scores of each item are summed to obtain a total crude score, which is then multiplied by 1.25 to obtain a standard score according to the Chinese norm. The higher the

standard score, the more severe the symptoms. The standard score for evaluating depression is 53, with a score below 53 indicating no depression, 53~62 indicating mild depression, 63~72 indicating moderate depression, and a score of 72 or above indicating severe depression [27].

**2.2.2 Body dissatisfaction.** This study evaluated body dissatisfaction using sex-adapted silhouettes for men and women [28], which contain a total of 15 body size figures. With the center of the middle figure extending symmetrically to the left and right scales, the body type progression chart scores present an hierarchical progression from the lowest-scoring (-7) obese body type to the highest-scoring (7) muscular body type. Respondents were asked to select their current body silhouette (Current Silhouettes, CS) and their ideal body silhouette (Ideal Silhouettes, IS), that is, the body type they would most like to be and which they perceived as most attractive to others. The resulting difference was body dissatisfaction (IS-CS). Based on these results, survey respondents were classified according to their body dissatisfaction: low dissatisfaction, $|IS, CS| \leq 1$; medium dissatisfaction, $2 \leq |IS\text{-}CS| \leq 4$; and high dissatisfaction, $|IS\text{-}CS| \geq 5$.

**2.2.3 Sleep.** The quality of respondents' sleep was evaluated using the Pittsburgh Sleep Quality Index (PSQI) scale [29]. The PSQI is a 19-item self-report questionnaire assessing sleep quality over the past month and yielding seven component scores: sleep latency, sleep duration, habitual sleep efficiency, sleep disturbance, use of sleep medication, daytime dysfunction, and overall sleep quality. The PSQI score is the sum of these components, with a score greater than 7 considered evidence of a sleep disorder, and higher scores indicating poorer sleep quality in the previous month.

**2.2.4 Physical activity level.** Respondents' physical activity level was evaluated using the Physical Activity Rating Scale [30]. This scale examines the amount of physical activity the respondent engaged in over the previous month by surveying the intensity, duration, and frequency of physical activity, with each item measured with a score ranging from 1 to 5. The following formula was used to calculate the total physical activity score: Exercise intensity × (Exercise time−1) × Frequency of exercise. Scores range from 0 to 100 and are classified as follows: low intensity (≤19 points), medium intensity (20~42 points), and high intensity (≥43 points).

## 2.3 Statistical analysis

Data were analyzed using IBM SPSS Statistics 26. This study conducted two independent sample t-tests to test for gender differences in respondents' height, weight, body fat, muscle mass, depression scores, sleep scores, physical activity scores, and body dissatisfaction scores. This study also used a Pearson's test to compare differences between male and female respondents in terms of sleep quality, level of physical activity, and level of body dissatisfaction. In conducting multiple regression analysis, this study used respondent depression scores as the dependent variable and sleep, physical activity, and body dissatisfaction scores as predictor variables. Variables were selected using a stepwise increasing and decreasing method with a threshold p-value of 0.20, and calculated using likelihood ratio tests. All items were tested for normality and skewness, and the data were within acceptable limits; P < 0.05 was considered a statistically significant difference.

## 2.4 Sample size estimation

The sample size for the study was determined using the G*Power calculator 3.1.9.7 (Franz Faul et al., Universität Kiel, Germany, http://www.gpower.hhu.de/). Based on α = 0.05, 1−β = 0.90, four tested predictors (SDS score, body dissatisfaction, physical activity, and sleep duration), and three covariates (age, sex, and BMI), we calculated the sample size to be 27, 59, 430 if the

effect size $f^2$ equaled 0.35 (large), 0.15 (medium), and 0.02 (small), respectively. Furthermore, assuming a 20% dropout rate, the total number was estimated as 34–538. To ensure the desired power, the sample size was increased to 1,355, producing an actual valid sample size of 1,258, which was much larger than the estimated size even when using a small effect size $f^2$.

## 2.5 Ethics approval and consent to participate

This study was approved by the ethics committee of the Gannan Medical University, China, No: 2021110. This study was conducted according to the guidelines in the Declaration of Helsinki, and all study participants provided informed consent, agreeing to the required measurement and survey completion procedures. All methods were performed in accordance with the relevant guidelines and regulations.

## 3. Results

### 3.1 Depression in university students

As Fig 1 shows, 319 survey respondents experienced depression. Of these, 235 (18.7%) experienced mild depression, 79 (6.3%) experienced moderate depression, and 5 (0.4%) experienced severe depression. Results thus indicate an overall depressive mood detection rate of 25.4%.

### 3.2 Differences between male and female students

As Table 1 and Fig 2 shows, there were statistically significant (P < 0.01) differences between male and female respondents in terms of BMI, body fat, muscle mass, depression score, PSQI score, physical activity, and body dissatisfaction. Men had significantly higher BMI, muscle mass, and physical activity scores than women, while women had significantly higher body fat percentage, depression scores, and PSQI scores than men. Female respondents scored higher on the sleep scale than males (P < 0.01).

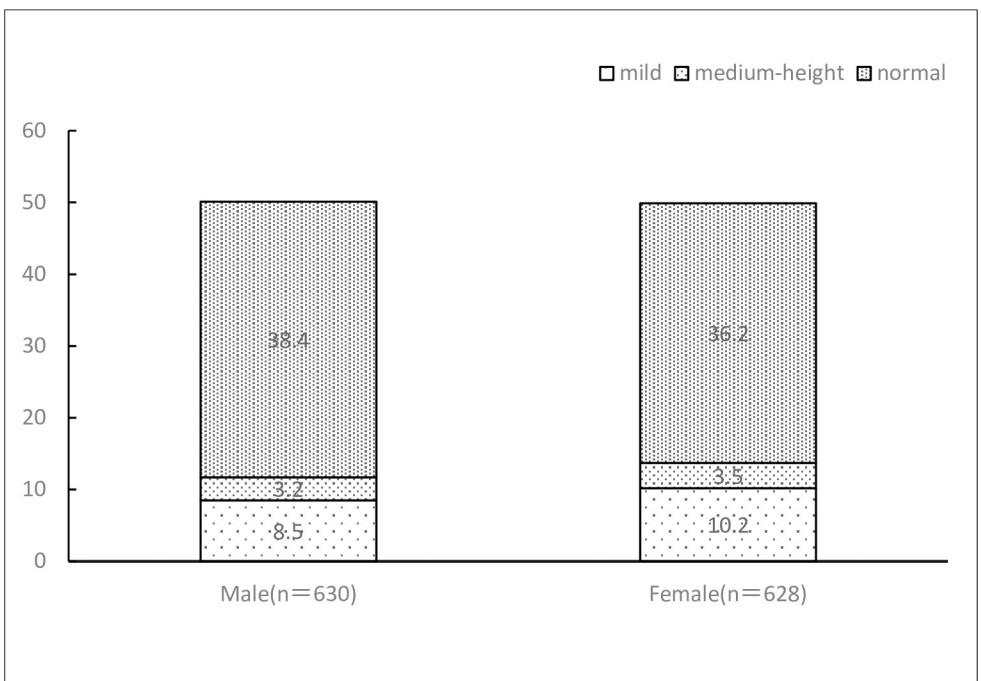

**Fig 1. Detection of depression among university students (n = 1,258).**

**Table 1. Comparison of values between boys and girls.**

|  | Mean ± SD or n (%) | | P |
| --- | --- | --- | --- |
|  | Male (n = 630) | Female (n = 628) |  |
| BMI (kg/m2) | 22.1±3.7 | 21.2±3.1 | <0.01 |
| Fat% | 16.1±7.2 | 27.3±6.6 | <0.01 |
| Muscle mass (g) | 50.4±8.3 | 35.9±4.7 | <0.01 |
| PSQI score | 5.03±2.29 | 5.48±2.17 | <0.01 |
| Physical activity score | 21.4±20.8 | 12.3±16.3 | <0.01 |
| Physical activity category |  |  |  |
| None exercise | 60(9) | 89(14) | <0.05 |
| Low exercise | 319(51) | 435(69) | <0.01 |
| Medium exercise | 143(23) | 57(9) | <0.01 |
| High exercise | 108(17) | 47(8) | <0.01 |
| Body dissatisfaction | 1.63±1.26 | 1.24±1.09 | <0.01 |
| Body dissatisfaction category |  |  |  |
| Low | 318 (51) | 412 (65) | <0.01 |
| Medium | 292 (46) | 212 (34) | <0.01 |
| High | 20 (3) | 4 (1) | <0.01 |

Note. BMI, body mass index. The significance of differences between male and female students was determined using t-tests (for quantitative variables) or Pearson analyses (qualitative variables).

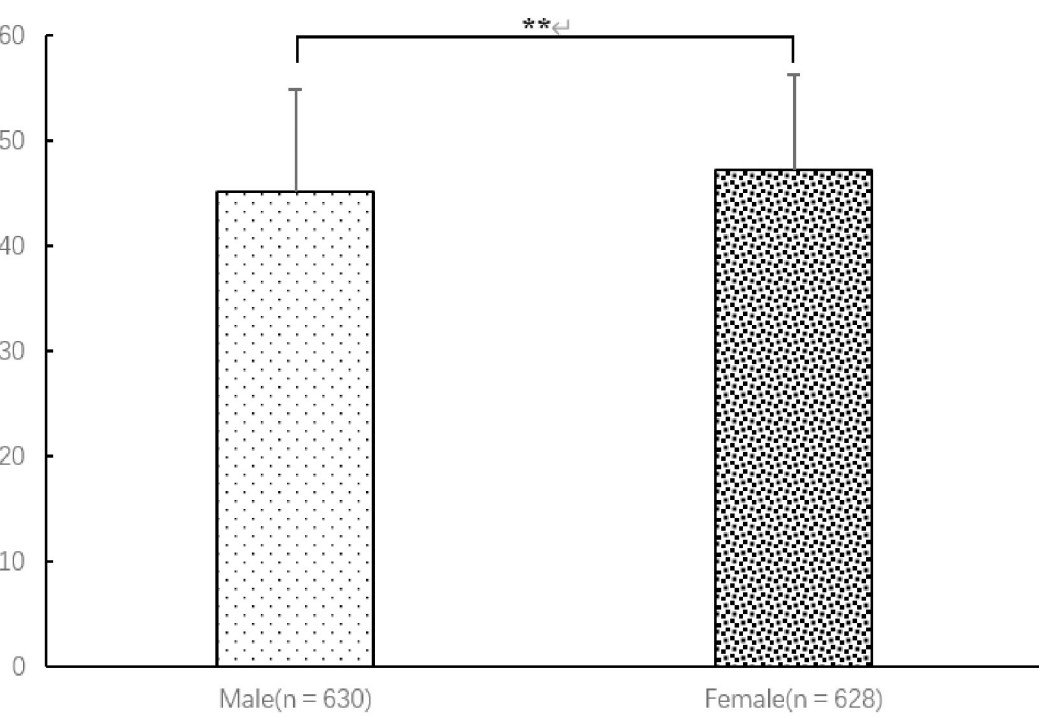

** p < 0.01

**Fig 2. Depression scores of male and female respondents.**

**Table 2. Distribution of university students' physical activity.**

| | Number of people | | Total | constituent ratio(%) |
|---|---|---|---|---|
| | **Male** | **Female** | | |
| Low | 379 | 524 | 903 | 71.8 |
| Medium | 143 | 57 | 200 | 15.9 |
| High | 108 | 47 | 155 | 12.3 |
| Total | 630 | 628 | 1258 | 100 |

### 3.3 Physical activity

As Table 2 shows, of the total sample, 71.8% exercised at low intensity (42% male, 58% female), 15.9% at medium intensity (72% male, 28% female), and 12.3% at high intensity (70% male, 30% female). Approximately 83% of female respondents exercised at a low intensity, while approximately 40% of male respondents exercised at a moderate to high intensity.

### 3.4 Factors affecting depression levels

This study conducted a stepwise multiple regression analysis of male and female respondents using total depression score as the dependent variable and the physical activity, sleep, and body dissatisfaction scores as independent variables. As the results presented in Table 3 show, the PSQI score (β = 1.768, P < 0.01) and physical activity score (β = -0.048, P < 0.01) were significant predictors of depression among male respondents, while the PSQI score (β = 1.743, P < 0.01) and body dissatisfaction score (β = 0.917, P < 0.01) were significant predictors of depression among female respondents.

## 4. Discussion

### 4.1 Depression levels and gender differences among university students in southern China

The outbreak of COVID-19 brought about considerable inconvenience to people's lives, with workers unable to work normally and students prevented from starting school on time [7]. The various control and prevention policies invariably led to anxiety among some students, resulting in an increase in depression levels in the population [7]. Indeed, a survey involving more than 1,400 undergraduates at 40 US medical schools showed a 70% higher rate of depression among students during the outbreak compared with previous studies [31]. In the Chinese context, a meta-analysis of research published on depression among university students in

**Table 3. Factors associated with depression in undergraduate students.**

| | β | t | VIF | p |
|---|---|---|---|---|
| **Male[†]** | | | | |
| PSQI score | 1.768 | 11.358 | 1.006 | <0.01 |
| Physical activity | -0.048 | -2.794 | 1.006 | <0.01 |
| **Female[‡]** | | | | |
| PSQI score | 1.743 | 11.708 | 1.005 | <0.01 |
| Body dissatisfaction | 0.917 | 3.095 | 1.005 | <0.01 |

Note. VIF = variance inflation factor

[†]: $R^2$: 0.187; P < 0.01; DW:1.988

[‡]: $R^2$: 0.196; P < 0.01; DW:1.905

China between 1997 and 2015, which identified some 39 studies examining a total of 32,694 university students, revealed an overall prevalence rate of depression among university students in China of 23.8% [32]. Conducted in 2020, a large-scale cross-sectional study of more than 700,000 university students enrolled in 108 universities in the Guangdong and Jiangxi Provinces identified a depression rate of 21.1% [33]. Focusing on the pandemic period, this study found a high level of depression among respondents at the University of Southern China the identified rate of 25.4% considerably higher than the results of pre-pandemic research. Significantly, research indicates that the global estimates of the overall prevalence of depression have increased over time [34]. In this respect, it is likely that the prevalence of depression among Chinese university students will continue growing. Therefore, it is particularly important to design and implement depression improvement and prevention strategies for university students as early as possible.

Research conducted both before and after the pandemic observed higher rates of depression among female university students [35, 36]. In line with previous findings, the results of this study also found a higher rate of depression among female respondents. Physiological differences may explain the gender differences in depression levels among university students [37]. Some studies suggest that the biological differences between men and women may be reflected in emotions and behavior, with women likely to be more emotional in the face of challenging and uncertain situations [37]. During the pandemic, students' inability to participate in social activities and attend school regularly significantly impacted their academic progress and exacerbated any emotional problems [38]. Research indicates that female students were more susceptible to tension contagion, more emotional and moody, and more prone to negative emotions than their male peers, with a significant increase in the incidence of depression among females [39]. As such, sex differences were likely magnified during the pandemic.

This gendered difference in depression levels may also have been exacerbated by patriarchal issues in Chinese society. In this regard, studies have shown that parents-to-be tend to prefer boys to girls [40], gender inequality persists in childhood, parents tend to spend more energy on their sons than their daughters, and the majority of families with sons report a happier life [41]. Studies also indicate that women tend to be more sensitive because they are not taken seriously [42], are more prone to sentimentality, and prefer keeping negative emotions bottled up rather than letting them out like men. As women are more susceptible to stress and pain than men, they may experience greater levels of sadness and anxiety [43], as well as elevated levels of depression.

## 4.2 The effect of body dissatisfaction on depression

According to the results of this study, body dissatisfaction was only a significant predictor of depression levels (β = 0.917, P < 0.01) among female university students (Table 3). Body dissatisfaction is prevalent in the population because of negative preconceptions associated with the failure to achieve or maintain a socially desirable body image [44]. Research has also shown that body dissatisfaction is negatively associated with self-esteem and life satisfaction and positively associated with depressed mood in women [45]. Likewise, this study did not observe any association between body dissatisfaction and depression levels among male respondents (Table 3). It is worth noting that body dissatisfaction in women is no longer restricted to obese women, but seen in women with normal and higher levels of body dissatisfaction [46].

This difference can be attributed to factors such as social acceptance and social culture. For example, slim women have a greater advantage on the marriage market, with a slim body believed to increase the happiness of couples in their marriages [47]. Consequently, the female

population tends to identify more strongly with the image of slim women, and thus spend more time thinking about their body image and are at greater risk of developing psychological problems related to body dissatisfaction [37, 48]. People with body dissatisfaction typically try to change their body image [49], often resorting to diet pills, active emetics, prolonged periods of not eating, and laxatives [50, 51]. This type of dieting can be extremely taxing on the body and mind, is prone to rebound, and may increase the risk of depression [52].

## 4.3 The effect of sleep on depression

The WHO recommends that adults achieve an average of seven to nine hours of sleep per night [53]. Approximately 26% of the respondents in this study failed to meet this standard, with 16.5% were considered to have a sleep disorder (PSQI > 7). Rather than the recommended amount of seven hours, 97% of those who did not meet this sleep standard slept six hours on average. Approximately 33% of those with a depressive mood did not meet the WHO's recommended sleep standard. This aligns with the findings of a meta-analysis of 89 studies (n = 1,441,828) published in 2021, which showed that the prevalence of depressive symptoms and sleep disorders had increased to 34% and 33%, respectively—both figures higher than evaluations of similar populations before the pandemic [4].

A study of 859 university students found that students with mental disorders had significantly lower overall sleep quality than those without mental disorders [54]. Similarly, a study conducted between October 2005 and April 2007 on the quality of sleep of approximately 4,513 students at a large state university found that students who did not sleep well tended to experience mental disorders like depression [55]. Scholars have also shown that sleep disturbance is an effective predictor of depression [56]. Similarly, this study's multiple regression analysis of factors influencing depression levels found that sleep quality significantly predicted depression levels in both men and women and that improvements in sleep quality were significantly associated with reduced depression levels (Table 3). As such, the findings of this study support the results of prior studies.

This study also found that female respondents had significantly higher sleep scores than male respondents (Table 1). The difference in sleep quality between male and female respondents was statistically significant (P < 0.05), indicating that women's sleep quality was significantly worse than that of men. This findings is consistent with the results of studies showing that women tend to have more sleep disorders and poorer sleep quality than men [57]. Poor sleep quality may be a significant contributor to the sex differences in the prevalence of depression. The vast majority of university students live on campus and not in a controlled environment with an enforced curfew. Consequently, they tend to, lack self-restraint, are easily influenced by those around them, and are likely to continue chatting and surfing the Internet when they should be sleeping [55]. Therefore, raising students' awareness of the importance and value of sleep is key to preventing and improving depression levels among university students in the post-pandemic period.

## 4.4 The effect of physical activity on depression

Analysis of the relationship between basic physical activity and poor psychological status among 500 university students found that sedentary students scored higher than active students on factors such as stress, anger, fatigue, depression, and panic, but scored lower on energy and self-esteem. Simply put, the higher the physical activity index, the better the mental health, interpersonal, and emotional status of the university students[58]. Conversely, the less physical activity, the more pronounced the unhealthy psychological status of the university students [58]. In this regard, the results of this study show that exercise activity was only a

significant predictor of depression among male respondents, who reported fewer depressive symptoms and higher levels of exercise (Table 3).

Moreover, of the 319 individuals who experienced depression in this study, 262 (82%) engaged in low-intensity exercise (Table 2). This finding is consistent with related studies suggesting that a lack of exercise is significantly associated with increased levels of depression [43]. For instance, a 2012 study using a random stratified sample of more than 5,000 students from 58 universities reported that 87.4% of students said they felt better after exercising, while 81.5% felt that their current exercise was insufficient [59]. A meta-study demonstrated that regular exercise improves sleep quality [60]. According to the results of a study of 1,143 university students in Beijing, rigorous physical activity and quality sleep helped regulate and reduce depressive symptoms in university students, with a depression detection rate of 37% for men and 43% for women [61]. Meanwhile, a cross-sectional study surveying over 1,000 university students in Wuhan in 2011, found that the prevalence of poor sleep quality and depression were 17.7% and 10.6%, respectively [62]. The study also found that high levels of exercise were associated with a significantly lower prevalence of depression and high sleep quality among university students, both independently and interactively [62]. Significantly, regular exercise can improve one's self-concept and thus one's mental health [63]. These results suggest that active participation in physical activity and the development of a moderate and regular exercise program can significantly reduce depression levels.

This study did not observe an association between exercise activity and depression in females (Table 3). While exercise levels were relatively lower among female respondents, this study found that female respondents who reported lower levels of exercise also reported higher levels of body dissatisfaction. In contrast, this study found that the level of exercise activity in male respondents significantly predicted their level of depression (Table 3). Nevertheless, the lack of such an association between the two variables among female respondents in this study echoes the results of several previous studies [64, 65]. Interestingly, a study of 85 undergraduate students at a Japanese university in 2021, found a positive correlation between depression and physical activity levels: the higher the intensity of physical activity, the more likely one was to be depressed. This study concluded that physical activity leads to sleep deprivation and increased daytime sleepiness, causing a deterioration in mental health [66]. Indeed, while current research generally agrees that regular exercise is effective for improving sleep quality, some suggest that intense exercise is an important factor in exercise-related insomnia [67]. Therefore, while exercise can help reduce depression levels, we must be careful not to overdo it, as this can lead to exercise dependency, which can have negative effects.

These findings notwithstanding, this study has several limitations. First, the study sample comprised just over 1,000 participants, all of whom were university students aged 18–22 years. Furthermore, as this is a cross-sectional study, no causal inferences can be drawn. Second, the study only examined college students living in southern China and is not representative of the country as a whole. The survey was conducted during the COVID-19 pandemic, when China implemented a strict "zero Covid" policy and experienced the longest and strictest lockdowns, which could skew the results. Therefore, the results may not reflect the overall relationship among body dissatisfaction, sleep, exercise, and depression in college students. Third, the results were based on a self-reported survey and may be subject to bias and inaccuracy. Fourth, considering the ongoing COVID-19 epidemic, the results of this study can provide a theoretical basis for formulating strategies to improve sleep, exercise behavior, and depression levels of college students. This study did not investigate what specific actions they took when feeling low-an oversight, we hope to address in future research. Future studies should consider recruiting a wider range of respondents from different regions and types of universities to

obtain a wider range of results on the relationship between body imagery, physical activity, sleep, and depression among university students.

The results of this preliminary analysis showed that the depression levels of university students in southern China were significantly higher among females than among males. We found a correlation between body dissatisfaction, physical activity, and sleep and depression. The findings of this study indicate that the role of body dissatisfaction in causing depression may be underestimated. Although improving body satisfaction may directly influence depression levels, increasing physical activity and improving sleep quality may also be beneficial. Therefore, reducing body dissatisfaction has significant potential for preventing and ameliorating depression.

## Author Contributions

**Data curation:** Bang Liu, Ming Hao.

**Formal analysis:** Bang Liu, Ming Hao.

**Funding acquisition:** Ming Hao.

**Investigation:** Bang Liu, Xuesheng Liu, Lin Zou, Jun Hu, Yueming Wang, Ming Hao.

**Methodology:** Bang Liu, Ming Hao.

**Writing – original draft:** Bang Liu, Ming Hao.

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
