## [Decision Letter · Decision Letter 0]

31 Jul 2023

PONE-D-23-11540The effects of body dissatisfaction, sleep duration, and exercise habits on the mental health of university students in southern China during COVID-19PLOS ONE

Dear Dr. Hao,

Thank you for submitting your manuscript to PLOS ONE. After careful consideration, we feel that it has merit but does not fully meet PLOS ONE’s publication criteria as it currently stands. Therefore, we invite you to submit a revised version of the manuscript that addresses the points raised during the review process. Your manuscript has been evaluated by two reviewers, and their comments are appended below. Please ensure you address each of the reviewers' comments when revising your manuscript. Please note that one or more reviewers has recommended that you cite specific previously published works. As always, we recommend that you please review and evaluate the requested works to determine whether they are relevant and should be cited. It is not a requirement to cite these works. We appreciate your attention to this request.

We look forward to receiving your revised manuscript.

Kind regards,

Hugh Cowley

Staff Editor

PLOS ONE

Journal Requirements:

"This study was supported by the Starting Research Fund from the Gannan 

Medical University, QD202121 Research start-up funds for high-level talents, Project number :2044 and Ganzhou city federation of social science 2021-028-0538."

"This study was supported by the Starting Research Fund from the Gannan 

Medical University, QD202121 Research start-up funds for high-level talents, Project number :2044 and Ganzhou city federation of social science 2021-028-0538."

Reviewers' comments:

Reviewer's Responses to Questions

**Comments to the Author**

1. Is the manuscript technically sound, and do the data support the conclusions?

Reviewer #1: Yes

Reviewer #2: Yes

2. Has the statistical analysis been performed appropriately and rigorously? 

Reviewer #1: Yes

Reviewer #2: Yes

3. Have the authors made all data underlying the findings in their manuscript fully available?

Reviewer #1: Yes

Reviewer #2: Yes

4. Is the manuscript presented in an intelligible fashion and written in standard English?

Reviewer #1: Yes

Reviewer #2: Yes

5. Review Comments to the Author

Reviewer #1: To begin, I would want to express my gratitude to the authors for all of the hard work they put into conducting and presenting this fascinating study. I am aware that the process of peer review can be difficult; however, I want you to be assured that my objective is to provide you with constructive input that will assist in enhancing the quality of your article and will contribute to the overall success of the project.

Abstract:

The abstract describes a study on southern Chinese university students' mental health and COVID-19. The study examines body dissatisfaction, sleep length, and exercise habits with student depression. The abstract describes the study's goals, methodologies, and findings.

The abstract summarizes study findings. Depression was detected at 25.4%, with greater rates in women. The abstract also lists male and female depression predictors. Sleep quality (PSQI score) and physical activity were significant predictors for males, while sleep quality and body dissatisfaction were significant for women.

The abstract concludes that university students have mental health issues during the COVID-19 pandemic. It recommends boosting student sleep quality and physical activity.

The abstract summarizes the study well. It explains the setting, objectives, methods, significant findings, and mental health intervention recommendations. The abstract summarizes the findings on university students' mental health during the COVID-19 pandemic in southern China.

Overall it is good but some minor scientific flaws in the abstract:

The abstract does not specify whether the investigation was cross-sectional or longitudinal. Understanding the associations' causation requires this knowledge.

The abstract does not describe the sample strategy used to recruit 1,258 university students. How participants were selected and whether the sample is representative of the population of interest are crucial.

Causal assertions without evidence: The abstract states that body dissatisfaction, physical activity, and sleep predict depression. The abstract does not show that these variables cause depression. Establishing causality requires good study design and analysis.

The abstract cites gathering data on respondents' anthropometric measurements, body dissatisfaction, eating habits, sleep status, physical activity levels, and depression levels, but it does not specify the measurement instruments utilized. To ensure accuracy, incorporate measurement instrument validity and reliability information.

Statistics ignorance: The abstract lists beta coefficients (β) and p-values but not statistical methodologies. To assess results robustness, describe statistical analyses and control variables.

The abstract does not highlight study limitations. To properly understand results, sampling bias, self-reporting biases, and generalizability must be considered.

Unfounded advice: The abstract advises university students to increase sleep and exercise. How these recommendations were developed from the study findings is unclear without explicit intervention plans or evidence-based therapies.

Introduction:

1. The introduction mentions statistics such as the rise in depression and anxiety disorders during the first year of COVID-19 and the number of individuals with depression in 2018. However, neither the sources nor the context is provided for these statistics. It is essential to clarify the origin of the data and its relevance to the current investigation.

2. The transition between discussing the global impact of COVID-19 and focusing on university students in the Jiangxi Province of China is abrupt and lacks a clear explanation. The connection between the general populace and college students should be better explained and justified.

3. The introduction contains unsupported generalizations regarding the relationship between body dissatisfaction and depression, the impact of sleep disorders on mental health, and the beneficial effects of exercise on depression. However, neither specific references nor empirical evidence support these claims. It is essential to provide evidence to substantiate such claims.

4. Absence of specific research questions or objectives: In the introduction, neither the specific research questions nor the objectives of the study are explicitly stated. It is essential to clearly articulate the purpose and objectives of the research in order to provide a road map for the study and determine the reader's expectations.

5. Insufficient explanation of the significance of the study: Although the introduction mentions the goal of reducing the risk of depression among university students and providing a scientific basis for early prevention and intervention, it does not explain the significance of the study in the context of mental health research or the potential impact of the findings.

Methodology:

1. The anthropometric measurements section mentions the use of a "height ruler (model)" and a "body composition analyzer (model)" without supplying specific model information. To ensure the reproducibility of a study, it is essential to provide explicit model names and specifications.

2. The section on the SDS scale, which is used to evaluate the severity of depression, mentions that scores are multiplied by 1.25 to obtain a standard score according to Chinese norms. However, the rationale or foundation for this factor is not explained. It is essential to provide a detailed explanation of the study's scoring system.

3. Classification of body dissatisfaction is missing an explanation. The methodology section describes how respondents were categorized as having low, moderate, or high dissatisfaction based on the difference between their ideal and actual body silhouettes. However, neither the classification criteria nor the thresholds for classifying respondents into these categories are provided. It is essential to clarify how the classification was determined and which values correspond to each category.

4. The statistical analysis section mentions that parametric tests were conducted under the assumption that the data followed a normal distribution, but provides no additional information. However, no information is provided regarding the verification or evaluation of this assumption. It is essential to describe any analyses or procedures used to confirm the data's normality assumption.

5. Not justified threshold p-value for variable selection: In the methodology section, it is stated that variables for the multiple regression analysis were selected using an increasing and decreasing method with a p-value threshold of 0.20. However, no explanation or justification is provided for the selection of this particular p-value threshold. It is essential to provide a justification or reference for the chosen threshold.

6. Insufficient information on sample size calculation: The methodology section contains no information on how the sample size was calculated. It is essential to describe the rationale or methodology used to calculate the sample size, ensuring that it is sufficient for the statistical analyses and research objectives.

Discussion:

I read the content provided and did not notice any obvious mistakes in writing. I may, however, make the following improvements and clarifications:

1. The text discusses a variety of sample sizes for numerous research; however it does not consistently present the sample sizes for each individual finding. The sample sizes for each study or analysis referenced should be stated in detail.

2. Absence of citations for particular studies the manuscript makes numerous references to studies without properly citing them. To make it easier for readers to find and cite the original research, it is crucial to include complete citations for all studies that are discussed.

3. Generalizations without specific support: Some claims are made in general without referencing any specific studies or supporting data. It would be helpful to give specific references or evidence to back up these claims, particularly when talking about societal and gender-based disparities.

4. Lack of discussion of restrictions: The document only briefly refers to restrictions; it does not go into further detail. More in-depth discussion of the potential drawbacks of the study's design, data collection techniques, and generalizability of the results would be beneficial.

5. The manuscript makes a case for the need for further study but does not offer any concrete research questions or recommendations for additional studies. Based on the current findings, specific recommendations for further research would be beneficial.

6. In order to improve the scientific rigor and clarity of the findings given, the publication would benefit from providing more particular information, identifying specific studies, and addressing limitations.

Reviewer #2: To the authors of this paper:

This manuscript was well thought out and written. Your methodology and analyses fit the scope of the paper and were applicable to answering the research aims of the paper. A few edits are needed.

1) Line 154

-Please provided the entire name of the SDS scale since this is the first time it was introduced within the paper

2) Discussion (Section 4.2 The effect on body dissatisfaction on depression)

-A recent paper published (Differences in Perceived Stress and Depression among Weight (Dis)Satisfied Midwestern College Students during COVID-19>>>https://www.mdpi.com/2673-8112/3/5/56)

might be a good paper to reference related to your discussion of females spending more time dealing with psychological issues related to body image and body dissatisfaction [Lines 329-332]).

3) References

-References are inconsistently formatted.

-Journal names are in all caps for some, and formatted differently in others.

-Please double check all and correct

6. PLOS authors have the option to publish the peer review history of their article (what does this mean?). If published, this will include your full peer review and any attached files.

Reviewer #1: No

Reviewer #2: No

---

## [Author Response · Author response to Decision Letter 0]

7 Sep 2023

Thank you for the opportunity to review this manuscript which has aim to assess the association between body dissatisfaction, sleep length, and exercise habits with student depression among university students at a unique moment in the lives of everyone around the world as was the covid-19 pandemic.

Reviewer 1:

Abstract

Question 1 The abstract does not specify whether the investigation was cross-sectional or longitudinal. Understanding the associations' causation requires this knowledge.

Answer 1: Thank you for your comments. This study is a cross-sectional study, as we have explained in the abstract section (Line 33). 

Question 2 The abstract does not describe the sample strategy used to recruit 1,258 university students. How participants were selected and whether the sample is representative of the population of interest are crucial. 

Answer 2: Thank you for your comments. We added the sample strategy for recruiting these 1258 college students in the abstract section (Line 32-33). 

Question 3 Causal assertions without evidence: The abstract states that body dissatisfaction, physical activity, and sleep predict depression. The abstract does not show that these variables cause depression. Establishing causality requires good study design and analysis. 

Answer 3: Thank you for your comments. I have revised the problems you mentioned in the conclusion of the abstract (Line 42-44). 

Question 4 The abstract cites gathering data on respondents' anthropometric measurements, body dissatisfaction, eating habits, sleep status, physical activity levels, and depression levels, but it does not specify the measurement instruments utilized. To ensure accuracy, incorporate measurement instrument validity and reliability information.

Answer 4: Thank you for your comments. I very much agree with your opinion, but due to the limited space of the abstract part, I have no way to add the specific relevant content to the abstract, but I have added the specific content to the method part.

: the instruments used to measure anthropometric data (Line 150-154), body dissatisfaction (Line 171-172), sleep status (Line 186-187), physical activity level (Line 196-197), and depression level of the respondents (Line 158-159). 

Question 5 Statistics ignorance: The abstract lists beta coefficients (β) and p-values but not statistical methodologies. To assess results robustness, describe statistical analyses and control variables. 

Answer 5: Thank you for your comments. We have added statistical methods to the abstract section (Line 37). 

Question 6 The abstract does not highlight study limitations. To properly understand results, sampling bias, self-reporting biases, and generalizability must be considered. 

Answer 6: Thank you for your comments. We have added information about the limitations of our study at the end of the abstract (Line 44-47). 

Question 7 Unfounded advice: The abstract advises university students to increase sleep and exercise. How these recommendations were developed from the study findings is unclear without explicit intervention plans or evidence-based therapies.

Answer 7: Thank you for your comments. We have removed the advice and changed it to the following:“Results indicate the possibility of alleviating depression among university students by improving their body dissatisfaction, physical activity, and sleep (Line 42-44).”

Introduction:

Question 1 The introduction mentions statistics such as the rise in depression and anxiety disorders during the first year of COVID-19 and the number of individuals with depression in 2018. However, neither the sources nor the context is provided for these statistics. It is essential to clarify the origin of the data and its relevance to the current investigation.

Answer 1: Thank you for your comments. We have added relevant references to the article on statistics such as the rise in depression and anxiety (Line 63-64) in the first year of COVID-19 and the number of people with depression (Line 75-76) in 2018. The relevance of this data to the current investigation is highlighted. 

Question 2 The transition between discussing the global impact of COVID-19 and focusing on university students in the Jiangxi Province of China is abrupt and lacks a clear explanation. The connection between the general populace and college students should be better explained and justified. 

Answer 2: Thank you for your comments. Introduction the background of the article (Line 79-92), we mentioned in the new college students during the period of pandemic depression detection rate is significantly higher than the general population, and psychological health problems of college students are vulnerable groups, so college students as the research object of this research has better representative. 

Question 3 The introduction contains unsupported generalizations regarding the relationship between body dissatisfaction and depression, the impact of sleep disorders on mental health, and the beneficial effects of exercise on depression. However, neither specific references nor empirical evidence support these claims. It is essential to provide evidence to substantiate such claims.

Answer 3: Thank you for your comments. Body dissatisfaction is common among college students. However, in the context of the pandemic, the relationship between body dissatisfaction and the development of depression is still unclear. This article is to further explore the relationship between the two. We have added relevant references to the impact of sleep disturbances on mental health (Line 105-106) and the beneficial effects of exercise on depression (Line 117-119). 

Question 4 Absence of specific research questions or objectives: In the introduction, neither the specific research questions nor the objectives of the study are explicitly stated. It is essential to clearly articulate the purpose and objectives of the research in order to provide a road map for the study and determine the reader's expectations. 

Answer 4: Thank you for your comments. At the end of the introduction, we clarified that the research question of this paper was to explore the relationship between body image, sleep, physical activity and depression level among college students in Jiangxi Province (Line 129-131), and the goal was to explore the possibility of improving body dissatisfaction on depression in college students. This study hopes to provide a scientific basis for early prevention and intervention of depression in college students, thereby addressing the large population burden of mental disorders (Line 131-136). 

Question 5 Insufficient explanation of the significance of the study: Although the introduction mentions the goal of reducing the risk of depression among university students and providing a scientific basis for early prevention and intervention, it does not explain the significance of the study in the context of mental health research or the potential impact of the findings. 

Answer 5: Thank you for your comments. I have explained the research significance of this paper in detail in the introduction (Line 131-136).

Methodology:

Question 1 The anthropometric measurements section mentions the use of a "height ruler (model)" and a "body composition analyzer (model)" without supplying specific model information. To ensure the reproducibility of a study, it is essential to provide explicit model names and specifications.

Answer 1: Thank you for your comments. We specify in the article method section the anthropometric data of respondents used in concrete model name and specifications (Line 150-154). “Height was measured using a height ruler (Seca 213, Germany) with an accuracy of 0.1 cm, while weight (0.1 kg), body fat percentage, muscle mass (kg), and bone mass (kg) were measured using a body composition analyzer (Tanita BC-610, Japan). Body mass index (kg/m2) was calculated using the height and weight measurements”.

Question 2 The section on the SDS scale, which is used to evaluate the severity of depression, mentions that scores are multiplied by 1.25 to obtain a standard score according to Chinese norms. However, the rationale or foundation for this factor is not explained. It is essential to provide a detailed explanation of the study's scoring system.

Answer 2: Thank you for your comments. We have explained the calculation principle of this score in the article of the source of the original scale, and the calculation of the score is described as follows: "The depression is evaluated by the depression severity index, and the depression severity index = the cumulative score of each item /80 (the highest total score). The index ranges from 0.25 to 1.00, with higher indices indicating more severe depression." Formula to rounded, this paper will finally multiplied by one hundred, made the last decimal into integer convenient, also is the sum of each item total score multiplied by 1.25. The final classification criteria were no depression at 53, mild depression at 53 to 62, moderate depression at 63 to 72, and severe depression at 72 or higher. Relevant reference has been added in the article (Line 161-168). The previous study changed the 80-point system to one-hundred-mark system. This method has been widely promoted and is well applied in southern China (Wu Chunmei et al, 2023 doi:10.3389/fnut.2023.1103724).

Question 3 Classification of body dissatisfaction is missing an explanation. The methodology section describes how respondents were categorized as having low, moderate, or high dissatisfaction based on the difference between their ideal and actual body silhouettes. However, neither the classification criteria nor the thresholds for classifying respondents into these categories are provided. It is essential to clarify how the classification was determined and which values correspond to each category.

Answer 3: Thank you for your comments. We have added details on the classification of body dissatisfaction in the manuscript and added references to support it (Line 180-183) as follows: 

 “Based on these results, survey respondents were classified according to their body dissatisfaction: low dissatisfaction, �IS-CS� ≤ 1; medium dissatisfaction, 2 ≤ �IS-CS� ≤ 4; and high dissatisfaction, �IS-CS� ≥ 5.”.

Question 4 The statistical analysis section mentions that parametric tests were conducted under the assumption that the data followed a normal distribution, but provides no additional information. However, no information is provided regarding the verification or evaluation of this assumption. It is essential to describe any analyses or procedures used to confirm the data's normality assumption. 

Answer 4: Thank you for your comments. All items were tested for normality and skewness, and the data were within acceptable limits (Line 217-219). 

Question 5 Not justified threshold p-value for variable selection: In the methodology section, it is stated that variables for the multiple regression analysis were selected using an increasing and decreasing method with a p-value threshold of 0.20. However, no explanation or justification is provided for the selection of this particular p-value threshold. It is essential to provide a justification or reference for the chosen threshold.

Answer 5: Thank you for your comments. We determined the threshold of a p value of 0.2 by referring to articles in the same type and in the same region and consulting statistical experts. (Wu Chunmei et al, 2023 doi:10.3389/fnut.2023.1103724)

Question 6 Insufficient information on sample size calculation: The methodology section contains no information on how the sample size was calculated. It is essential to describe the rationale or methodology used to calculate the sample size, ensuring that it is sufficient for the statistical analyses and research objectives. 

Answer 6: Thank you for your comments. We have provided detailed principles and methods for the calculation of sample size in this study at the end of the Methods section of this article (Line 222-232). 

Discussion:

Question 1 The text discusses a variety of sample sizes for numerous research; however it does not consistently present the sample sizes for each individual finding. The sample sizes for each study or analysis referenced should be stated in detail. 

Answer 1: Thank you for your comments. We have fully supplemented the sample size for each reference.

Question 2 Absence of citations for particular studies the manuscript makes numerous references to studies without properly citing them. To make it easier for readers to find and cite the original research, it is crucial to include complete citations for all studies that are discussed.

Answer 2: Thank you for your comments. I have been added to increase the reference of related literature.

Question 3 Generalizations without specific support: Some claims are made in general without referencing any specific studies or supporting data. It would be helpful to give specific references or evidence to back up these claims, particularly when talking about societal and gender-based disparities. 

Answer 3: Thank you for your comments. We have added specific references in the relevant places in the article.

Question 4 Lack of discussion of restrictions: The document only briefly refers to restrictions; it does not go into further detail. More in-depth discussion of the potential drawbacks of the study's design, data collection techniques, and generalizability of the results would be beneficial.

Answer 4: Thank you for your comments. We have supplemented this in detail in the section on the limitations of this article at the end of the article (Line 455-474).

Question 5 The manuscript makes a case for the need for further study but does not offer any concrete research questions or recommendations for additional studies. Based on the current findings, specific recommendations for further research would be beneficial. 

Answer 5: Thank you for your comments. We have refined specific research questions and additional research proposals (Line 470-474).

Question 6 In order to improve the scientific rigor and clarity of the findings given, the publication would benefit from providing more particular information, identifying specific studies, and addressing limitations. 

Answer 6: Thank you for your comments. We have perfected the conclusion of the scientific rigour and clarity (Line 475-483).

Reviewer 2:

Question 1 Please provided the entire name of the SDS scale since this is the first time it was introduced within the paper. 

Answer 1: Thank you for your comments. We have supplemented the full SDS scale in the article (Line 159). 

Question 2 A recent paper published (Differences in Perceived Stress and Depression among Weight (Dis)Satisfied Midwestern College Students during COVID-19>>>https://www.mdpi.com/2673-8112/3/5/56)

might be a good paper to reference related to your discussion of females spending more time dealing with psychological issues related to body image and body dissatisfaction.

Answer 2: Thank you for your comments. After carefully reading the article you suggested, the content is indeed suitable, and we have added this reference in the article (Line 355-358).

Question 3 References are inconsistently formatted. Journal names are in all caps for some, and formatted differently in others.

Answer 3: Thank you for your comments. All formats of the articles have been unified according to the format requirements of the journal.

---

## [Decision Letter · Decision Letter 1]

2 Oct 2023

The effects of body dissatisfaction, sleep duration, and exercise habits on the mental health of university students in southern China during COVID-19

PONE-D-23-11540R1

Dear Dr. Hao,

We’re pleased to inform you that your manuscript has been judged scientifically suitable for publication and will be formally accepted for publication once it meets all outstanding technical requirements.

Kind regards,

Guglielmo Campus, Ph.D DDS

Academic Editor

PLOS ONE

Additional Editor Comments (optional):

Reviewers' comments:

Reviewer's Responses to Questions

**Comments to the Author**

1. If the authors have adequately addressed your comments raised in a previous round of review and you feel that this manuscript is now acceptable for publication, you may indicate that here to bypass the “Comments to the Author” section, enter your conflict of interest statement in the “Confidential to Editor” section, and submit your "Accept" recommendation.

Reviewer #1: All comments have been addressed

Reviewer #2: All comments have been addressed

2. Is the manuscript technically sound, and do the data support the conclusions?

Reviewer #1: Yes

Reviewer #2: Yes

3. Has the statistical analysis been performed appropriately and rigorously? 

Reviewer #1: Yes

Reviewer #2: Yes

4. Have the authors made all data underlying the findings in their manuscript fully available?

Reviewer #1: Yes

Reviewer #2: Yes

5. Is the manuscript presented in an intelligible fashion and written in standard English?

Reviewer #1: Yes

Reviewer #2: Yes

6. Review Comments to the Author

Reviewer #1: Authors have addressed all the issues that I have raised in their previous submission. I appreciate their efforts and attempts to their hard works.

Reviewer #2: (No Response)

7. PLOS authors have the option to publish the peer review history of their article (what does this mean?). If published, this will include your full peer review and any attached files.

Reviewer #1: **Yes: **Rubayet Shafin

Reviewer #2: No

---

## [Editor Report · Acceptance letter]

4 Oct 2023

PONE-D-23-11540R1 

The effects of body dissatisfaction, sleep duration, and exercise habits on the mental health of university students in southern China during COVID-19 

Dear Dr. Hao:

I'm pleased to inform you that your manuscript has been deemed suitable for publication in PLOS ONE. Congratulations! Your manuscript is now with our production department. 

Kind regards, 

on behalf of

Prof. Dr. Guglielmo Campus 

Academic Editor

PLOS ONE